# Protein Levels of Pro-Inflammatory Cytokines and Chemokines as Biomarkers of *Mycobacterium bovis* Infection and BCG Vaccination in Cattle

**DOI:** 10.3390/pathogens11070738

**Published:** 2022-06-29

**Authors:** Hamza Khalid, Anouk van Hooij, Timothy K. Connelley, Annemieke Geluk, Jayne C. Hope

**Affiliations:** 1Division of Infection and Immunity, The Roslin Institute, University of Edinburgh, Easter Bush, Edinburgh EH25 9RG, UK; timothy.connelley@roslin.ed.ac.uk; 2Center for Inflammation Research, The Queen’s Medical Research Institute, Edinburgh BioQuarter, 47 Little France Crescent, Edinburgh EH16 4TJ, UK; 3Department of Infectious Diseases, Leiden University Medical Center, 2333 ZA Leiden, The Netherlands; a.van_hooij@lumc.nl (A.v.H.); a.geluk@lumc.nl (A.G.)

**Keywords:** BCG, biomarkers, bovine tuberculosis, cytokines, chemokines, diagnostics, pro-inflammatory

## Abstract

Bovine tuberculosis (bTB), caused by *Mycobacterium bovis*, is a globally prevalent infectious disease with significant animal welfare and economic impact. Difficulties in implementing test-and-slaughter measures in low- and middle-income countries (LMICs) and the underperformance of the current diagnostics establish a clear need to develop improved diagnostics. Adaptive immunity biomarkers other than IFNγ could be useful as suggested by various gene expression studies; however, a comprehensive assessment at the protein level is lacking. Here, we screened a range of chemokines and cytokines for their potential as biomarkers in samples from *M. bovis* experimentally challenged or naive animals. Although serum concentrations for most proteins were low, the pro-inflammatory markers, IL-2, CXCL-9, IP-10 and CCL4, in addition to IFNγ, were found to be significantly elevated in bovine tuberculin (PPDb)-stimulated whole blood supernatants. Further assessment of these molecules in BCG-vaccinated with or without subsequent *M. bovis* challenge or naive animals revealed that PPDb-specific IL-2 and IP-10, in addition to IFNγ, could discriminate naive and BCG-vaccinated from *M. bovis* challenged animals. Moreover, these proteins, along with CCL4, showed DIVA potential, i.e., enabling differentiation of *M. bovis*-infected animals from BCG-vaccinated animals. Combined analysis of cytokines and chemokines could also accurately identify *M. bovis* infection with strong correlations observed between PPDb-specific IFNγ, IL-2 and IP-10 levels. This provides proof of concept for utilizing multiple biomarker signatures for discrimination of animals with respect to *M. bovis* infection or BCG vaccination status.

## 1. Introduction

Bovine tuberculosis (bTB) is a major livestock infectious disease with severe implications for animal welfare, trade and economics. *Mycobacterium bovis* (*M. bovis*), the causative agent of TB in cattle, belongs to the mycobacterium tuberculosis complex (MTBC) [1]. On an individual animal level, bTB is a chronic, often subclinical disease, resulting in the formation of microscopic or characteristic macroscopic “tubercles” which are mostly pulmonary (lungs and associated lymph nodes) or, more rarely, extra-pulmonary (intestines, liver, spleen, etc.) [2]. As is the case for human TB, transmission among cattle herds occurs most likely via the aerosol route (through pulmonary bTB). Global estimates indicate that approximately 50 million cattle are infected and that the annual economic loss incurred is USD 3 billion [3]. The funds to control bTB in the UK alone for the decades 2014–2024 exceeded GBP 1 billion [4].

Upon infection, *M. bovis* mycobacteria initially interact with resident macrophages in the lung tissues which, in turn, become activated and act to control the infection via Th1-type cytokines, mainly IFNγ. In addition, a number of pro- and anti-inflammatory cascade cytokines and chemokines are released with roles in activation and recruitment of immune cells at the site of infection to mount an effective adaptive immune response. As a result, granulomas are formed which help to limit spread but also result in tissue damage such as caseation [5,6]. Humoral (antibody) responses, if present, usually are detectable at later stages of the disease and correlate to increased bacterial burden and pathology [7].

The ante-mortem diagnosis of bTB relies on the tuberculin skin test (TST). The TST involves measuring the delayed-type hypersensitivity reaction (DTH) at the site of bovine tuberculin (PPDb) administration, which is used either alone (single intradermal test—SIT) or alongside avian tuberculin (PPDa), usually in the cervical region (single intradermal comparative cervical tuberculin Test; SICCT). This comparative assessment allows for the distinction between cattle sensitized by environmental mycobacteria and those infected with *M. bovis*. The skin thickness at the injection site is monitored and compared 72 h following inoculation [8] with absolute increases in PPDb-specific DTH being indicative of bTB. While local interpretations may differ, within Great Britain, under standard conditions cattle are classified as bTB skin test reactors (positive for *M. bovis)* if the increase in the thickness at the PPDb injection site is 4 mm greater than the increase observed at the site of PPDa administration. In cattle herds with repeated identification of bTB reactors, animals exhibiting a PPDb–PPDa increase that is >2 mm are defined as bTB reactors [9]. The skin test is most widely used, but it is liable to operator influences, co-infections, etc., rendering compromised sensitivity and specificity. The interferon gamma release assay (IGRA) [10], by contrast, allows comparatively early detection following *M. bovis* infection but requires sophisticated laboratory equipment and that blood is stimulated within 8 h after sampling [11], making it less applicable in remote settings [12,13]. The sensitivities for SICCT and IGRA are in the ranges of 52–100% and 73–100%, while the specificities are in the ranges of 78–99% and 85–99%, respectively, resulting in false negative and false positive animals [3,14]. The underperformance of TST and IGRA in field situations and the presence of wildlife reservoirs of *M. bovis* (e.g., the Eurasian badger in the UK) mean that control of bTB is challenging.

Test-and-slaughter measures are implemented for bTB control in a number of developed countries, but this approach is not economically feasible in low- and middle-income countries (LMICs). Thus, other control measures are required such as vaccination; Bacille Calmette Guerin (BCG), a live attenuated strain of *M. bovis*, is the only licensed vaccine against TB in humans. The effectiveness of BCG vaccination in cattle for preventing severe disease/granulomatous tissue lesions is well described in experimental studies and protective efficacy, although variable, has been shown in field trials [15]. The biggest challenge in adopting BCG as a control measure is the sensitization of vaccinated animals, which then test positive in the TST and IGRA. This inability to differentiate *M. bovis*-infected animals from vaccinated animals (DIVA) hampers the deployment of BCG vaccination as a control measure. Considerable progress has been made since the genome sequencing of *M. bovis* [16] to overcome this by identifying and testing antigens that are present in *M. bovis* but not present in BCG (e.g., 6 kD early secretory antigen target-6 kDa (ESAT-6), culture filtrate protein-10 kDa (CFP-10)). Promising antigen combinations are now under small-scale testing and field evaluation in the TST [17,18].

Considering the caveats of TST and IGRA coupled with the complex subclinical dynamics of bTB and limited information on endpoints referring to exposure, latent infection and active disease, there has been an interest in finding additional and/or alternative biomarkers. These may aid in detecting animals that are otherwise missed in the currently used routine tests. Since cell-mediated immune responses are vital for protection against mycobacteria, cytokines and chemokines are lead candidates for biomarker discovery. Studies from human mycobacterial diseases (i.e., TB and leprosy) have demonstrated the feasibility and added value of host biomarker signatures composed of multiple host proteins for diagnosing patients with different disease phenotypes [19,20]. An important aspect of this approach is the potential of simultaneous and quantitative measurement of multiple biomarkers on a single lateral flow strip providing point-of-care (POC) solutions [21].

Bovine TB studies have also demonstrated the potential of achieving enhanced diagnostic sensitivity and specificity by determining multiple proteins’ levels and incorporating them as an additional readout to IFNγ [22,23,24,25]. Differential gene expression in *M. bovis*-infected animals compared to naive controls has been reported for interleukin 1 beta (IL-1β) [22,26], IL-2 [27], IL-4 [28], IL-8 [29,30], IL-10 [31,32], IL-17A [33,34], IL-22 [35], TNFα [36] and the chemokines CXCL-9 (C-X-C- motif ligand-9) and interferon gamma inducible protein 10 (IP-10) [23,37,38,39]. The majority of cattle studies report mRNA expression differences in peripheral blood mononuclear cells (PBMCs) after stimulation with mycobacterial antigens or, less commonly, from lymph node tissues. These differences in mRNA expression can significantly discriminate experimentally *M. bovis*-infected animals from healthy controls. Although mRNA expression differences are a robust measure of elucidating immune-pathological profiles of diseases, a more practical approach would involve a comprehensive assessment and characterization of blood-derived proteins. We propose that a comprehensive host protein-based biomarker profile detectable in serum or stimulated whole blood supernatants that can differentiate animals with respect to infection and vaccination status would be useful to improve diagnostic sensitivity and specificity for bTB control.

In this study, we analyzed nine host proteins in bTB naive, BCG-vaccinated, BCG-vaccinated-*M. bovis* challenged and *M. bovis* challenged animals. We provide evidence for the utility of additional protein measurements to discriminate between these groups and show that multiple assessments can provide added sensitivity and specificity.

## 2. Results

### 2.1. Cytokine and Chemokines Levels in Naive and M. bovis Challenged Animals

In order to determine biomarkers allowing detection of *M. bovis* infection, an initial assessment was performed using samples from *M. bovis* challenged (C) and naive cohort 1 animals (N). Significantly higher concentrations of PPDb-specific IFNγ, IL-2, IP-10, CXCL-9 and CCL4 were observed in stimulated whole blood supernatants of *M. bovis* challenged animals compared to the naive group (Figure 1A, Appendix A). In serum samples from the same cohort, concentrations of the host proteins tested were generally low, in agreement with our observations in human TB [40]. However, statistically significant differences in serum expression of the chemokines CXCL-9 and IP-10 were noted between the two groups (Figure 1A).

Assessment of IL-8, IL-17A, IL-1β and IL-6 demonstrated elevated PPDb-specific levels for these cytokines in the *M. bovis* challenged cohort, although statistical significance was not reached. Serum concentrations of IL-8, IL-17A, Il-1β and IL-6 were generally very low or not detectable (Figure 1B, Appendix A).

### 2.2. Comparison of Cytokine and Chemokine Levels between Naive, BCG-Vaccinated, BCG-Vaccinated-M. bovis Challenged and Animals Challenged with M. bovis Alone

The host proteins that showed significant differences in the initial assessment (Figure 1), i.e., IFNγ, IL-2, CXCL9, IP-10 and CCL4, were further evaluated in bTB naive (cohort 2, N), BCG-vaccinated (V), BCG-vaccinated and subsequently *M. bovis* challenged (V/C) and *M. bovis* challenged (C) animals (Figure 2).

In stimulated whole blood supernatants, PPDb-specific IFNγ, IL-2 and IP-10 levels were found to be significantly different in naive animals compared to *M. bovis* challenged animals (N vs. C; *p* < 0.001). Levels of PPDb-specific IFNγ, IL-2 and IP-10 were highly significantly different between animals that were *M. bovis* challenged compared to BCG-vaccinated animals (C vs. V; *p* < 0.001 for IFN-γ and IL-2; *p* = 0.001 for IP-10), demonstrating that these three host proteins have strong DIVA potential at the studied time points. PPDb-specific CCL4 also showed DIVA potential to differentiate *M. bovis* challenge from BCG vaccination, albeit with lower significance (C vs. V; *p* = 0.04); in contrast to CXCL9, no significant differences were observed in PPDb-stimulated whole blood supernatants between any of the groups.

When analyzing the responses in BCG-vaccinated animals that were subsequently challenged with *M. bovis*, significant differences were observed in PPDb-specific IL-2 compared to naive animals (V/C vs. N; *p* = 0.001). Finally, levels of PPDb-specific IP-10 were able to discriminate between BCG-vaccinated *M. bovis* challenged animals and those which were challenged with *M. bovis* alone (V/C vs. C; *p* = 0.03)

Evaluation of serum samples from the same animals (Figure 3) demonstrated significant differences in CXCL-9 levels allowing differentiation between *M. bovis* challenged animals from naive (C vs. N; *p* = 0.002), BCG-vaccinated (C vs. V; *p* = 0.02) and BCG-vaccinated/*M. bovis* challenged animals (C vs. V/C; *p* < 0.001). Although some differences were noted in IFNγ, IL-2 and CCL4 between groups, the levels of expression were generally very low compared to the PPDb-specific levels (Figure 2). For the challenged group, serum concentrations were found to be highly variable for the evaluated proteins. No significant differences were noted in serum IP-10 levels between groups.

### 2.3. Performance of Host Proteins to Discriminate between Naive and M. bovis Challenged Animals

Next, we compared the protein expression levels of the naive animals (n = 16) with *M. bovis* challenged animals (n = 9) to assess the discriminatory potential of the host proteins in PPDb-stimulated whole blood supernatants (Table 1a) and in serum (Table 1b). Receiver operator characteristic curve (ROC curve) analysis was performed and the area under the curve (AUC) was calculated. The cut-offs were determined based on the ROC curves using Youden’s Index. For PPDb-specific levels, IFNγ and IL-2 at respective cut-off levels of 1943 and 2666 pg/mL could discriminate the *M. bovis* challenged animals from naive animals with a sensitivity and specificity of 100%. PPDb-specific IP-10 and CXCL-9 at cut-offs of 25,846 and 1651 pg/mL, respectively, could discriminate between the two groups with 100% sensitivity, while the specificities at these cut-offs were found to be 94% and 81%, respectively. Finally, PPDb-specific CCL4 at a cut-off value of 21,840 pg/mL could discriminate *M. bovis* challenged animals from naive animals with a sensitivity of 89% and specificity of 69%. For serum samples, the measurement of CXCL9 was most effective allowing discrimination between the groups with a sensitivity of 78% and specificity of 100% at the cut-off of 6377 pg/mL. For the other evaluated proteins, the cut-off values were very low (IFNγ (76 pg/mL), IL-2 (170 pg/mL) and CCL4 (15 pg/mL)) and for IP-10, a specificity of only 50% was observed. The ROC curves for the concentrations of each cytokine in medium/nil and PPDb-stimulated, as well as PPDb minus medium stimulated whole blood supernatants, and serum concentrations with respective AUC are shown in Figure 4.

### 2.4. Multibiomarker Analysis and Correlation between Cytokine and Chemokine Levels to Accurately Discriminate between Groups of Animals

Next, we analyzed whether a multiprotein signature could provide additional sensitivity and specificity in discriminating among the tested cohorts compared to individual protein assessments. To achieve this, we assigned a NUM (number) score to each animal by qualitative stratification, i.e., the NUM score is the number of proteins that are detected above the cut-off value for each individual animal [21].

Analysis of multiple comparisons showed that the NUM score could significantly differentiate *M. bovis* challenged animals from naive animals (*p* < 0.001) and at NUM score cut-off value of three, the groups could be differentiated with 100% sensitivity and specificity (Figure 5, Appendix A). Similarly, BCG-vaccinated animals and those that were challenged with *M. bovis* post-BCG could also be discriminated from *M. bovis* challenged only animals (*p* = 0.001 and *p* = 0.007, respectively); however, no significant differences for these two groups were observed compared to the naive group (*p* > 0.99 and *p* = 0.56, respectively).

Next, we wanted to investigate whether any of the other tested proteins (potential disease biomarkers) levels were correlated with IFNγ levels, since IFNγ is a currently used diagnostic biomarker in IGRA testing for bovine TB. For this, we performed correlation analysis, which showed that PPDb-specific IFNγ levels were strongly correlated to IL-2 and IP-10 levels (Pearson correlation R values of 0.86 and 0.81, respectively). A weaker correlation was observed with PPDb-specific CXCL9 (R = 0.61) and CCL4 levels (R = 0.54) (Figure 5).

A similar analysis with serum concentrations was performed (Appendix A). Although the cut-offs for individual proteins were very low, a NUM score cut-off > 3 demonstrated potential to discriminate naive and infected animals with a sensitivity of 89% and specificity of 100% (AUC = 1 and *p* < 0.001; Appendix A). Serum CXCL-9 levels, which performed best individually (Figure 3), showed weak correlations with serum concentrations of the other four tested proteins (Appendix A).

### 2.5. Correlations of Host Protein Concentrations with Bacteriology Counts and Total Lesion Scores

Next, for all of the animals in the study that were *M. bovis* challenged, we evaluated whether levels of the evaluated host proteins correlated to the bacterial counts and lesion scores determined at necropsy. For PPDB-stimulated supernatants, the highest correlation was observed for IP-10 (Pearson correlation coefficient square (R^2^) value of 0.74 and 0.63, respectively, for bacterial counts and total lesion score, *p* < 0.001). For the other four evaluated proteins, we observed highly significant (*p* < 0.001) yet weak correlations for both bacterial counts and total lesion scores (Figure 6A,B). When analyzing serum concentrations (Figure 6C), CXCL9 levels showed significant positive correlation with bacterial counts (*p* < 0.001, R^2^ =0.80). Though significant (*p* = 0.02), serum IP-10 showed weak correlation with the bacterial counts (R^2^ = 0.25). For total lesion scores, the only significant correlation observed in the serum protein levels was with CXCL-9 (*p* < 0.001, R^2^ = 0.62).

## 3. Discussion

Previous work assessing immune responses to *M. bovis* has identified cytokine and chemokine as biomarkers associated with infection using various gene expression techniques (RNA sequencing, microarrays, etc.) followed by validation of differential mRNA expression by qPCR. However, comprehensive data on validation of the differential expression of cytokines and chemokines at the protein level is lacking for *M. bovis* infection in cattle. This is essential if improved and accessible diagnostic tests incorporating biomarkers are to be developed and used in the field. Here, we compared cytokine/chemokine profiles in naive or BCG-vaccinated calves with or without subsequent *M. bovis* infection. We demonstrated that measurement of additional biomarkers (other than IFNγ—the only cytokine measured in commercial BOVIGAM^®^) can not only detect *M. bovis* infection but can also differentiate infected animals from BCG-vaccinated animals, those exposed to *M. bovis* post-BCG vaccination and those which are neither vaccinated nor exposed to *M. bovis*.

As expected, higher IFNγ levels were detected in stimulated whole blood supernatants after *M. bovis* infection, whereas post-BCG vaccination its levels were low. This is in line with previous studies of the kinetics of PPDb-specific expression of IFNγ post-BCG, which wanes by week 12 post-vaccination [41]. Our observations for the chemokines CCL4, IL-8, CXCL-9 and IP-10 are generally in agreement with previous studies [22,42,43]. Chemokines play a role in attracting activated T cells to promote Th1 response [31]. Although chemokine markers are generally detected in much high levels compared to IFNγ, one potential issue with them is nonspecific release [43]. Coad et al., reported that IP-10 when measured as an additional readout alongside IFNγ was shown to identify additional animals which were confirmed to have bTB but did not produce IFNγ responses [23]. At the assessment time points in our study, IL-2 and IP-10, in addition to IFNγ, showed a promise as DIVA markers i.e., differentiating infected animals from vaccinated animals. Antigen specific IL-2 responses for DIVA potential have also been described for naturally infected cattle previously [27]. Antigen-specific IP-10 levels also showed good potential to differentiate BCG-vaccinated and *M. bovis* challenged (protected or partially protected) animals from those infected with *M. bovis* alone.

Here, examination of IL-8, IL-1β, IL-6 and IL-17A levels in stimulated whole blood and sera alike demonstrated no significant differences between *M. bovis*-infected and naive animals. This is in contrast to other studies where PPDb-specific IL-8 was shown to discriminate *M. bovis*-infected and naive cattle and showed good agreement with TST and IGRA results in a double-blind assessment [42]. IL-1β drives IFNγ-mediated response and was previously reported to be significantly elevated in natural TB TST reactors compared to controls [22], but our observations with samples from *M. bovis* experimentally challenged animals showed the expression of IL-1β to be low and insignificant. The differences between studies may reflect the kinetics of expression and the time points chosen for analysis. For example, gene expression levels for IL-17A are time-point dependent, and IL-17 has been described as both a marker of infection (in *M. bovis* challenge experiments) as well as a correlate of BCG-induced protective immunity in cattle BCG vaccination studies [33]. In our study, IL-17 levels were low, consistent with what has recently been reported for both antigen-stimulated blood and interstitial fluid samples [44]. Low or undetectable protein levels coupled with kinetic dependent correlations with infection or protection limit the utility of cytokines as biomarkers of diagnostic relevance. Differences in age, sex and species may also account for some of the reported variance in the concentrations of proteins.

Despite the limitations of relatively small sample sizes and single time-point analyses reported here, our data show promise for the approach involving detection of multiple biomarkers (PPDb-specific IL-2 and IP-10 in addition to IFNγ) for the detection of *M. bovis* infection. These three proteins and CCL4 have significant potential to act as DIVA diagnostic markers enabling discrimination of *M. bovis* infection from BCG vaccination. Assessment of multiple biomarkers is indicative of differentiating active TB from other respiratory diseases (ORDs) in humans [40]. Mycobacterial studies from multiple veterinary species also support this notion. In African buffaloes, determining IP-10 levels in parallel to IFNγ levels from QuantiFERON^®^ plasma proved more sensitive in identifying all *M. bovis* culture positive animals compared to SICCT and IGRA performed individually or in parallel [45]. Utilizing a feline cytokine multiplex assay, O’Halloran et al. [46] reported the potential of multicytokine profiling in discriminating mycobacteriosis in cats based on etiology (cats infected with *M. bovis* and *M. microti* versus those infected with MTBC). In addition, Jones et al. [22] showed the potential for enhanced sensitivity (without loss of specificity) by incorporating IL-1β as an additional readout in parallel to IFNγ in whole blood from natural TB reactors stimulated with ESAT6/CFP-10. An important consideration is the type of sample being analyzed particularly in situations where access to a laboratory for stimulating blood, within an appropriate time window, with defined antigens may be limited. Serum may be a preferable sample type; however, our findings indicate serum concentrations of the evaluated cytokine and chemokine markers, except CXCL-9, were low (i.e., IFNγ, IL-2 and CCL4) or extremely variable (i.e., IFNγ, IL-2, IP-10 and CCL4) between animals in the same test group. This variability did not correlate in our study with the extent of disease (as measured by bacterial counts and total lesion scores).

Comprehensive studies comparing potential protein biomarker levels in different bTB infection and vaccination phenotypes and across different time points and different stages of disease are limited. An unbiased approach to biomarker discovery would be optimal but is hampered by lack of widely available bovine reagents, particularly in validating promising candidates. Further considerations include assessment of samples from naturally infected animals and the effect of co-infections relevant to field diagnosis including the cofounding effects of infection with *M. avium paratuberculosis* (MAP), the causative agent of Johne’s disease, shown previously to alter cytokine expression profiles and the bias of the immune response. Co-infection with MAP interferes with diagnostic accuracy of bTB diagnostics and is an important confounder to account for [47]. A lack of diagnostic accuracy may reflect the use of tuberculins (PPD) as stimulation antigens [48]. Future studies could incorporate assessment following stimulation with *M. bovis* specific antigens (i.e., ESAT6/CFP-10) that are lacking from BCG.

Nevertheless, biomarkers in addition to IFNγ have the potential to provide improved sensitivity of diagnostics and could be applicable to both human and veterinary medicine. Bovine TB vaccinology and diagnostics has benefited greatly from advances in human research and vice versa, and a One Health approach to disease control may be beneficial. Recently, a novel multibiomarker lateral flow format providing the possibility to quantitatively determine levels of biomarkers for both humoral and cellular immunity has been developed for human mycobacterial diseases [21]. This could have great applicability in the context of bovine TB where immune dynamics alters as the disease progresses and could incorporate, as well as cytokines and chemokines, key serum proteins, e.g., CRP [42] and amine oxidase, complement component 5 and serotransferrin, which show promise in mycobacterial diagnostics [49].

In summary, our data provide evidence that multiplex analysis holds significant promise for DIVA diagnostics and warrants further study to develop rapid, user-friendly tests for sensitive and specific diagnosis of bovine TB.

## 4. Materials and Methods

### 4.1. Selection of Candidate Biomarkers

A total of nine cytokines and chemokines were evaluated in the initial assessment. The selection of candidate biomarkers was based on (i) differential gene/protein expression from previous bovine TB studies [22,26,27,29,30,33,37] and (ii) availability of bovine-specific reagents either in the form of ELISA kits or antibody pairs with recombinant standards known and validated to work in an ELISA format; IFNγ, IL-1β, IL-2, IL-8, IL-17A, CXCL-9 and IP-10 were selected. In addition, CCL4 (C-C motif ligand 4/macrophage inflammatory protein 1 beta (MIP-1β)) and IL-6 were included based on data from human studies and the availability of bovine specific reagents [19,50]. Cytokines were measured by enzyme linked immunosorbent assay (ELISA) (Appendix A).

### 4.2. Samples and Animals

All samples were taken under license from the UK Home Office according to ASPA guidelines and with ethical approval from local Animal Welfare and Ethical Review Boards. Blood samples were taken by venepuncture from the jugular vein into either sodium heparin (10 IU/mL) for antigen stimulation or into tubes with no anticoagulant for serum collection. Samples were aliquoted and frozen at −80 °C until ELISAs were performed.

*Naive animal cohort 1* (n = 6). For initial screening of cytokines and chemokines, blood samples were taken from female animals aged approximately 6–12 months from the University of Edinburgh dairy farm (certified bovine TB free for >10 years).

*Naive animal cohort 2* (n = 10). Male Holstein–Friesian calves aged less than 1 month from the Institute for Animal Health (IAH) Compton dairy herd (certified bovine TB free for >10 years at the time of study). Naive samples represent the week 0 time point in the experimental study [41] and were taken prior to BCG vaccination.

*BCG-vaccinated cohort* (n = 10). The same ten calves as above, aged less than 1 month at the week 0 time point of the study [41], were vaccinated subcutaneously with either BCG Pasteur or BCG Danish strain vaccine (approximately 2 × 10^6^ CFU per calf sourced from the Veterinary Laboratories Agency, Weybridge). Blood samples were taken at 12 weeks post-vaccination.

*BCG-vaccinated and then M. bovis challenged cohort* (n = 12). British Holstein–Friesian calves from the IAH herd aged less than 1 month were vaccinated subcutaneously with BCG Pasteur (~1 × 10^6^ CFU per calf). Approximately twelve weeks post-vaccination, calves were infected with *M. bovis* strain AF2122/97 as previously described [41,51,52] and by Hope et al. (manuscript under preparation). Blood samples were taken 12 weeks post-*M. bovis* challenge.

*M. bovis challenged cohort* (n = 9). British Holstein–Friesian calves from the IAH herd aged approximately six months were challenged with *M. bovis* as described above and blood samples taken at 12 weeks following infection as previously described [41,51,52] and by Hope et al., (manuscript under preparation).

#### Preparation of Samples for Analysis

*Serum:* Blood was collected without anticoagulant, allowed to clot for 4 h at room temperature and then placed overnight at 4 °C. The caps and clots were then removed, the collected sera was centrifuged at 1500× *g* for 15 min at 4 °C and stored at −80 °C.

*Stimulated Whole Blood Supernatants*: In parallel with serum preparation, blood samples were collected into sodium heparin and subsequently stimulated as described [41]. Briefly, heparinized whole blood aliquots were incubated with RPMI medium alone (unstimulated/nil) or stimulated with purified protein derivative of *M. bovis* (PPDb) obtained from Veterinary Laboratories Agency (VLA, Weybridge, UK) at a final concentration of 20 µg/mL diluted in RPMI media with 50 µg/mL gentamicin. Supernatants collected after centrifugation were stored at −20 °C. Naive whole blood supernatants (cohort 1) were generated in the same manner, except that the PPD preparations were sourced from Prionics, Lelystad, the Netherlands.

The bacteriology counts and total lesion scores at necropsy were determined as described previously [41], and the data were used to study their correlation with the levels of host proteins.

### 4.3. ELISA Protocol

Sandwich ELISA were performed for nine biomarkers following the respective manufacturer’s protocols (Appendix A). Briefly, 100 μL of diluted antibody was used to coat plates overnight at 4–8 °C. Plates were washed five times with wash buffer (PBS + 0.05% Tween-20) between all subsequent steps. Plates were blocked by adding 200 μL of respective blocking buffers and incubated for 1 h at room temperature. Next, 100 μL of standards/samples diluted in respective buffers were added in duplicate and incubated for 1 h at room temperature. Diluted detection antibody (100 μL per well) was added, and following 1 h incubation and washing, diluted Streptavidin–HRP (100 μL/well) was added for 1 h at room temperature. TMB substrate (3,3’,5,5’—tetramethylbenzidine) was added (100 μL per well) for 15 min in the dark and the color reaction was then stopped by adding 100 μL of stop solution (0.2 M sulphuric acid). The optical density (OD) values at 450 nm (and 550 nm for background correction) were measured using microplate reader (BioTek, Winooski, VT, USA). Each plate had a standard curve, a positive control (Concanavalin A or LPS-stimulated PBMC supernatants) and a negative blank control, and each sample was assayed in duplicate. The concentrations were determined by extrapolating the standard curve using the average of the background corrected OD values for each sample. The detection limit of each assay was determined as the value (pg- or ng/mL) calculated from the mean + 2SD of OD value from the blank wells [53].

### 4.4. Statistical Analysis

GraphPad Prism version 9.0 for Windows (GraphPad Software, San Diego, CA, USA) was used to perform Mann–Whitney U tests, Kruskal–Wallis with Dunn’s correction for multiple testing, plot receiver operating characteristic (ROC) curves, calculate the area under the curve (AUC) and to generate Pearson correlation coefficients. The optimal sensitivity and specificity were determined using the Youden’s index [54]. The statistical significance level used was *p* < 0.05. Concentrations are expressed in picograms per milliliter (pg/mL).

## Figures and Tables

**Figure 1 pathogens-11-00738-f001:**
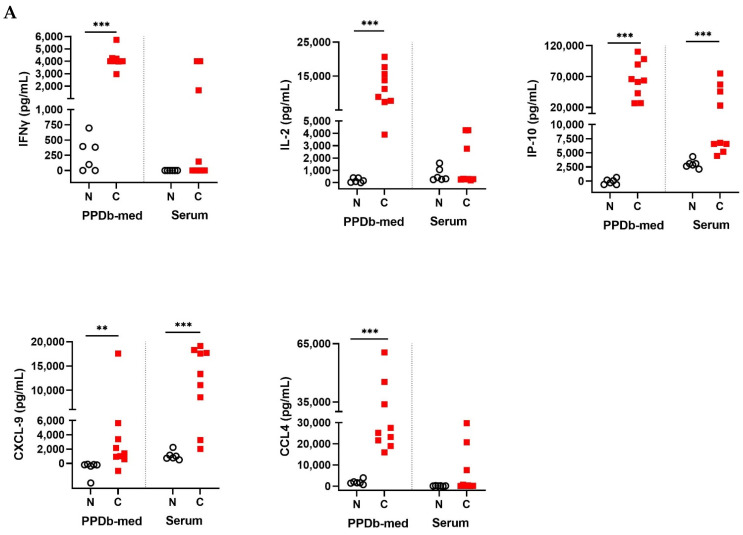
(**A**) Levels of IFNγ, IL-2, IP-10, CXCL-9 and CCL4 in stimulated whole blood supernatants and serum. Whole blood was stimulated for 24 h with PPDb or medium (med) alone and supernatants were assessed for the presence of cytokines and chemokines by ELISA. Shown are PPDb minus medium stimulated whole blood (PPDb-med) and serum concentrations expressed in pg/mL and compared using Mann–Whitney U test. N: Naive animals (cohort 1, n = 6, empty circles); C: *M. bovis* challenged animals sampled twelve weeks post-challenge (n = 9, red squares); ** *p* < 0.01; *** *p* < 0.001. (**B**) Protein levels of IL-1β, IL-6, IL-8 and IL-17A in stimulated whole blood supernatants and serum. Whole blood was stimulated for 24 h with PPDb or medium alone, and supernatants were assessed for the presence of cytokines and chemokines by ELISA. Shown are PPDb minus medium stimulated whole blood and serum concentrations expressed in pg/mL compared using Mann–Whitney U test. N: Naive animals (cohort 1, n = 6, empty circles); C: *M. bovis* challenged animals sampled twelve weeks post-challenge (n = 9, red squares).

**Figure 2 pathogens-11-00738-f002:**
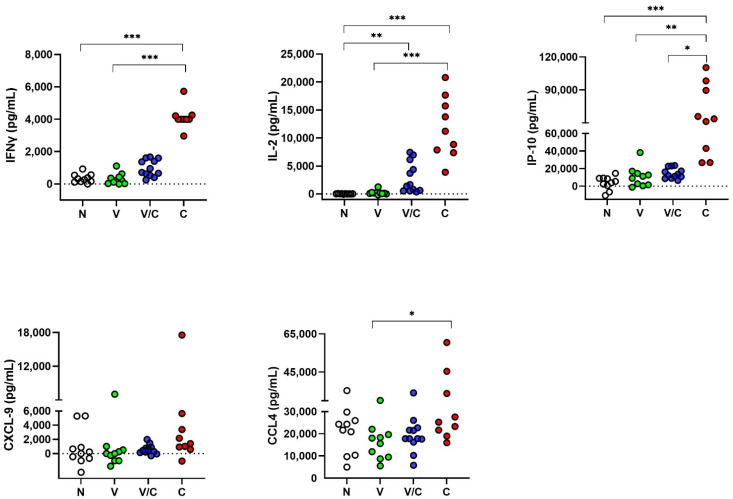
Protein levels of IFNγ, IL-2, IP-10, CXCL-9 and CCL4 in stimulated whole blood supernatants. Whole blood was stimulated for 24 h with PPDb or medium alone and supernatants were assessed for the presence of cytokines and chemokines by ELISA. Shown are PPDb minus medium stimulated whole blood concentrations expressed in pg/mL compared using the Kruskal–Wallis test with Dunn’s multiple comparison post-test. N: Naive animals (cohort 2, n = 10, empty circles); V: BCG-vaccinated animals (n = 10, green circles); V/C: BCG-vaccinated and *M. bovis* challenged (n = 12, blue circles); C: *M. bovis* challenged only (n = 9, red circles). * *p* < 0.05; ** *p* < 0.01; *** *p* < 0.001.

**Figure 3 pathogens-11-00738-f003:**
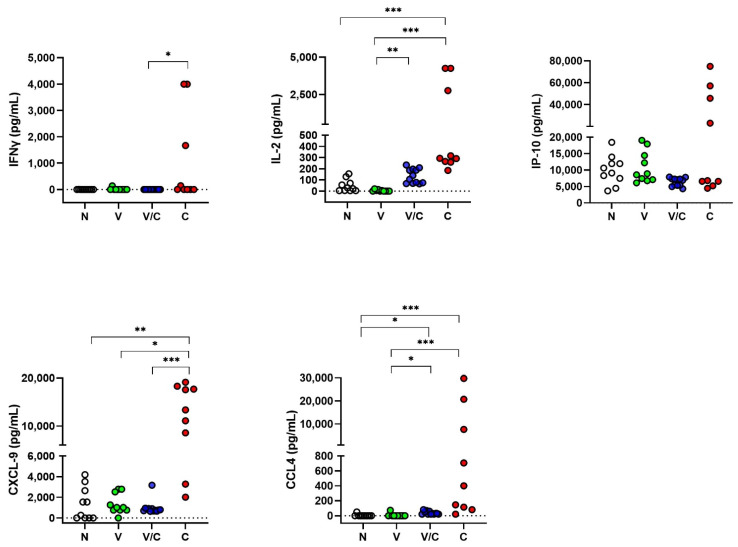
Protein levels of IFNγ, IL-2, IP-10, CCL4 and CXCL-9 in serum samples assessed by ELISA. Concentrations expressed in pg/mL compared using the Kruskal–Wallis test with Dunn’s multiple comparison post-test. N: Naive Animals (cohort 2, n = 10, empty circles); V: BCG-vaccinated animals (n = 10, green circles); V/C: BCG-vaccinated and *M. bovis* challenged (n = 12, blue circles); C: *M. bovis* challenged only (n = 9, red circles). * *p* < 0.05; ** *p* < 0.01; ***, *p* < 0.001.

**Figure 4 pathogens-11-00738-f004:**
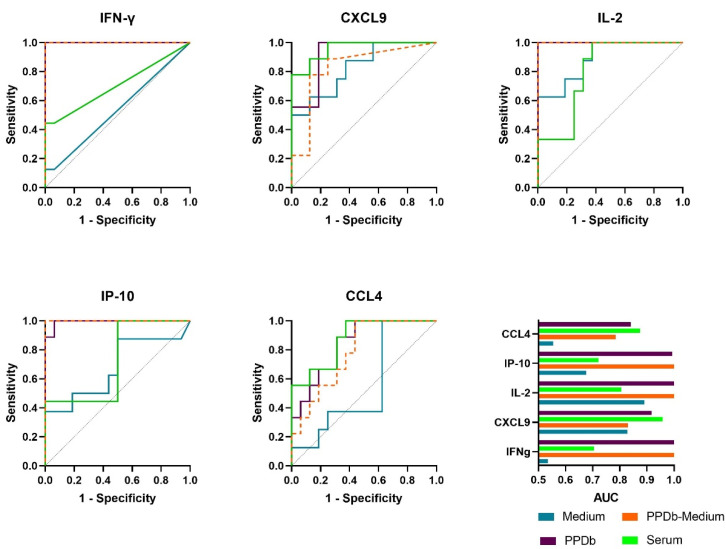
Receiver operator characteristic (ROC) curves of IFN-γ, CXCL9, IL-2, IP-10 and CCL4 detected in medium (blue), PPDB-stimulated whole blood supernatants (purple), PPDb-corrected for background (medium) values (orange) and serum (green) showing accuracy to differentiate *M. bovis* challenged animals (n = 9) from naive animals (n = 16). The bottom right panel provides an overview of the AUCs determined per protein in both serum and stimulated whole blood supernatant as determined in the depicted ROC curves.

**Figure 5 pathogens-11-00738-f005:**
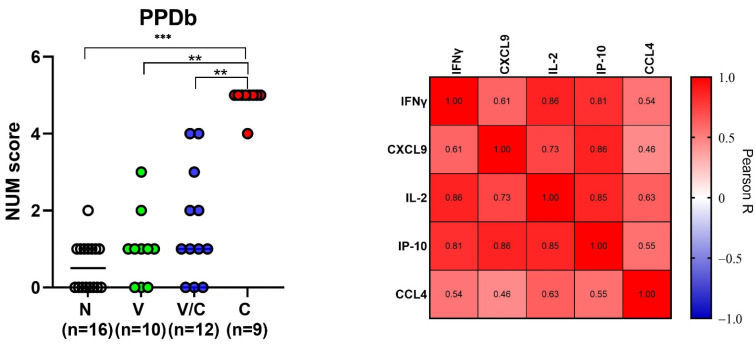
NUM score calculated based on IFN-γ, CXCL9, IL-2, IP-10 and CCL4 levels in PPDB-stimulated whole blood supernatants (cut-off values determined based on the optimal Youden’s index for naive (n = 16), BCG-vaccinated (n = 10) and *M. bovis* challenged animals with (n = 12) or without prior BCG vaccination (n = 9). The NUM score combines the results of the five proteins (*y*-axis), indicating the number of proteins with levels above a threshold as determined by the Youden’s index (Table 1). Group differences were determined using the Kruskall–Wallis test; the statistical significance level used was *p* < 0.05. ** *p* < 0.01, *** *p* < 0.001 (**left**). Heat map showing Pearson correlation among concentrations of the evaluated host proteins. The color corresponds to the Pearson R value as indicated in each square (**right**).

**Figure 6 pathogens-11-00738-f006:**
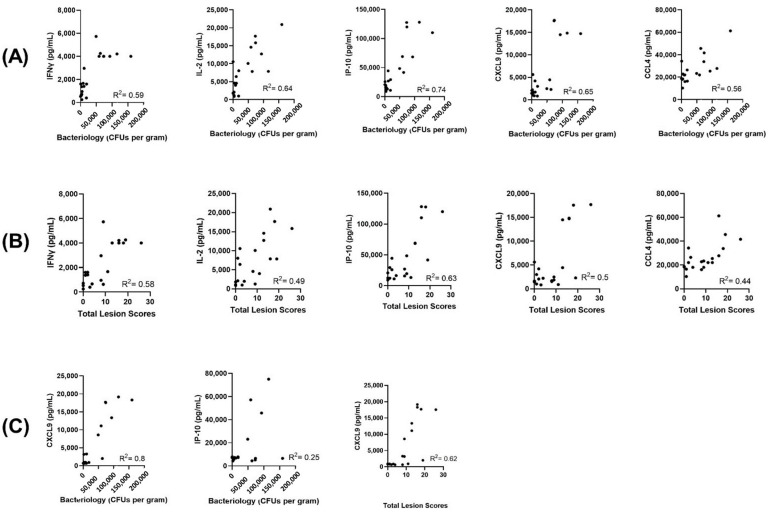
Correlations of host proteins’ concentrations (*x*-axis) with bacteriology counts (**A**) and total lesion scores (**B**) for PPDB-stimulated whole blood supernatants and (**C**) for serum at necropsy for *M. bovis* challenged animals. Bacterial counts and total lesions scores were determined as previously described [41]. R^2^ is the square of Pearson correlation coefficient.

**Table 1 pathogens-11-00738-t001:** Potential for IFNγ, IP-10, IL-2, CXCL9 and CCL4 to discriminate *M. bovis* infection.

**(a) Analysis by comparing levels in bovine tuberculin (PPDb)-stimulated whole blood supernatants**					
Analyte	Cut-off	Sensitivity (95% CI)	Specificity (95% CI)	AUC-ROC	*p*-value
IFNγ	>1943	100% (70.09 to 100.0)	100% (80.64 to 100.0)	1	<0.001
IP-10	>25,846	100% (70.09 to 100.0)	94% (71.67 to 99.68)	0.9931	<0.001
IL-2	>2666	100% (70.09 to 100.0)	100% (80.64 to 100.0)	1	<0.001
CXCL9	>1651	100% (70.09 to 100.0)	81% (56.99 to 93.41)	0.9167	<0.001
CCL4	>21,840	89% (56.50 to 99.43)	69% (44.40 to 85.84)	0.84	0.006
**(b) Analysis by comparing serum concentrations**					
Analyte	Cut-off	Sensitivity (95% CI)	Specificity	AUC-ROC	*p*-value
IFNγ	>76.00	44% (18.88 to 73.33)	100% (80.64 to 100.0)	0.7	0.09
IP-10	>4448	100% (70.09 to 100.0)	50% (28.0 to 72.0)	0.72	0.07
IL-2	>170.1	100% (70.09 to 100.0)	63% (38.64 to 81.52)	0.81	0.01
CXCL9	>6377	78% (45.26 to 96.05)	100% (80.64 to 100.0)	0.96	<0.001
CCL4	>15.00	100% (70.09 to 100.0)	63% (38.64 to 81.52)	0.88	0.002

Cut-off concentrations, sensitivities and specificities with 95% confidence intervals, AUC and *p*-values for cytokines in (a) whole blood supernatants stimulated with PPDb for 24 h and (b) serum samples.

## Data Availability

Not applicable.

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
