# Peer review of "Protein Levels of Pro-Inflammatory Cytokines and Chemokines as Biomarkers of Mycobacterium bovis Infection and BCG Vaccination in Cattle"

_pathogens, 2022, doi:10.3390/pathogens11070738_

Round 1

Reviewer 1 Report

The authors tested to see if cytokine/chemokine expression of PPDb-stimulated blood cells could be used for the diagnosis of bovine tuberculosis with the capability of differentiating the disease from vaccination. The results were promising and well presented. A couple of minor comments:

1.       It would be of interest to see if a machine learning classification method (I’d recommend ensemble learning or random forest) can improve the diagnostic accuracy. Since the stimulation of blood cells is a bottleneck for developing POC, I would be interested in seeing if such an approach allows for diagnosis (at least as a screening test) based on serum testing.

2.       In the last sentence of the discussion section, mentioning the possibility of POC test may not be justified as the test relies on the stimulation of blood cells.

Author Response

Reviewer 1

Comment: The authors tested to see if cytokine/chemokine expression of PPDb-stimulated blood cells could be used for the diagnosis of bovine tuberculosis with the capability of differentiating the disease from vaccination. The results were promising and well presented. A couple of minor comments:

It would be of interest to see if a machine learning classification method (I’d recommend ensemble learning or random forest) can improve the diagnostic accuracy. Since the stimulation of blood cells is a bottleneck for developing POC, I would be interested in seeing if such an approach allows for diagnosis (at least as a screening test) based on serum testing.

Response: Assessment of machine learning classification methods would be of interest to improve the diagnostic accuracy. This would be something we would actively consider when we have a larger sample size and we thank the reviewer for this suggestion. We agree that the stimulation of blood cells is a bottleneck for POC tests and we have (in line with the comment below), revised the text accordingly. The current aim of our studies is a rapid, user-friendly test that can be performed either directly after sampling or after (24h) incubation depending on the type of biomarker assessed in the test.

With respect to the use of serum for testing we have carried out analyses as presented in the manuscript (line 276-282 and Supplementary Figure 1. Although the cut-offs for individual proteins were very low, a NUM score cut-off > 3 demonstrated potential to discriminate naïve and infected animals with a sensitivity of 89% and specificity of 100% (AUC=1 and p<0.001; supplementary table 4). Serum CXCL-9 levels, which performed best individually (Figure 3) showed weak correlations with serum concentrations of the other four tested proteins (Supplementary Figure 1).

Comment: In the last sentence of the discussion section, mentioning the possibility of POC test may not be justified as the test relies on the stimulation of blood cells.

Response: We have modified the text accordingly (lines 396-397).

Reviewer 2 Report

The manuscript submitted by Hamza Khalid et al. entitled "Protein Levels of Proinflammatory Cytokines and Chemokines as Biomarkers of Mycobacterium bovis Infection and BCG Vaccination in Cattle" is about a very important cattle disease. Although the authors selected some of the previously reported cytokines, the possibility of these factors as biomarkers for the diagnosis of bovine tuberculosis was re-validated in rigorous animal studies. that's really good evidence for the diagnosis of bovine tuberculosis.

Author Response

Reviewer 2

Comment: The manuscript submitted by Hamza Khalid et al. entitled "Protein Levels of Proinflammatory Cytokines and Chemokines as Biomarkers of Mycobacterium bovis Infection and BCG Vaccination in Cattle" is about a very important cattle disease. Although the authors selected some of the previously reported cytokines, the possibility of these factors as biomarkers for the diagnosis of bovine tuberculosis was re-validated in rigorous animal studies. That’s really good evidence for the diagnosis of bovine tuberculosis.

Response: We thank the reviewer for their positive comments.

Reviewer 3 Report

This manuscript is based in the the overall need to improve dairy bovine tuberculosis diagnosis of tests with higher sensitivity by detection of truly infected animals by associations of immune biomarkers with standard testing, as animal infection often is missdiagnosed and has several important drawbacks due to animal conditions or false-negative interpretations, and/or micrbobial coinfections.

In general, this work with experimmentally infected animals with a short group of animals commplements what it has been over taken by several research groups and has been analyzed in several and individual efforts to optimize bovine tuberculosis diagnosis.

Although experimental models of infection could mimmic pathological findings of a true infection, authors should consider to improve research and originality within a natural setting in bovines with true cases of infection. Common observations in field examinations arise from asymptomatic animals with gross pathology at slaughter and importantly some with culture-negative to M. bovis, so, it has been stated that research and analysis of animal diagnosis would be improved by means of standard bovine Tb diagnosis and serological tests, as complexities arise in those animals with a wained immunity due to old age or immunusuporessive coinfections.

Results are generaly well described. One observation is that target immunomodulator molecules were individually measured by individual ELISA tests which means that micro variability of sample volumes and/or different time of sample manipulations and measurements could introduce variability in biomarker assessment, so a sole mutiplex array-based plataform could greatly increase research impact as has been used with this kind of research.

Figures are well organized and presented. The description of Figure 2 is not chronologically well indicated in the text.

As could be observed, the originality of the research findings in this work is to propose as DIVA molecules those best-expressed immunomodulators from the host, as DIVA reagents have been stated from microbiological origin to increase the sensitivity of in vitro diagnostic tests. However, as it is mentioned, findings first need to be extrapolated in field tests of natural infection. Discussion is well performed and bibliographically sustentated with other similar scientific works within the timeline, however, there are very recent reviews in this research line (https://doi.org/10.1128/IAI.00401-19; https://doi.org/10.1016/j.micres.2021.126853) as well that need to be considered to augment coverage of the final considerations of the conclusory text.

Author Response

Reviewer 3

Comment: This manuscript is based in the overall need to improve dairy bovine tuberculosis diagnosis of tests with higher sensitivity by detection of truly infected animals by associations of immune biomarkers with standard testing, as animal infection often is misdiagnosed and has several important drawbacks due to animal conditions or false-negative interpretations, and/or microbial coinfections.

In general, this work with experimentally infected animals with a short group of animals complements what it has been over taken by several research groups and has been analyzed in several and individual efforts to optimize bovine tuberculosis diagnosis.

Although experimental models of infection could mimic pathological findings of a true infection, authors should consider to improve research and originality within a natural setting in bovines with true cases of infection. Common observations in field examinations arise from asymptomatic animals with gross pathology at slaughter and importantly some with culture-negative to M. bovis, so, it has been stated that research and analysis of animal diagnosis would be improved by means of standard bovine Tb diagnosis and serological tests, as complexities arise in those animals with a waned immunity due to old age or immunosuppressive coinfections.

Response: We agree that we should assess naturally infected cattle and measure the impact of co-infections that could cause alterations in the immune response that then impact on diagnostic accuracy. We are in the process of collecting samples from field bTB reactors: these are tuberculin skin test positive animals and we will have access to data confirming their infection status including M. bovis culture data. We will also assess animals infected with Mycobacterium avium paratuberculosis/Johne’s disease: these will have confirmed diagnostic tests (antibody and/or qPCR). This will be important for determining the field sensitivity and specificity for both single ELISAs and multiplex assays. The results from these field validation studies will be published in due course.

Comment: Results are generally well described. One observation is that target immunomodulator molecules were individually measured by individual ELISA tests which means that micro variability of sample volumes and/or different time of sample manipulations and measurements could introduce variability in biomarker assessment, so a sole multiplex array-based platform could greatly increase research impact as has been used with this kind of research.

Response: We recognise that individual measurements and variability can impact on biomarker assessment. As we state in the manuscript, now that we have identified biomarkers that (jointly) improve the sensitivity, specificity and which can be used for DIVA diagnostics we will aim to develop multiplex lateral flow assays as reported by Van Hooij et al, 2021 https://doi.org/10.1016/j.isci.2020.102006.

Comment: Figures are well organized and presented. The description of Figure 2 is not chronologically well indicated in the text.

Response: We have added a citation to Figure 2 in the text (line 167).

Comment: As could be observed, the originality of the research findings in this work is to propose as DIVA molecules those best-expressed immunomodulators from the host, as DIVA reagents have been stated from microbiological origin to increase the sensitivity of in vitro diagnostic tests. However, as it is mentioned, findings first need to be extrapolated in field tests of natural infection. Discussion is well performed and bibliographically sustentated with other similar scientific works within the timeline, however, there are very recent reviews in this research line (https://doi.org/10.1128/IAI.00401-19; https://doi.org/10.1016/j.micres.2021.126853) as well that need to be considered to augment coverage of the final considerations of the conclusory text.

Response: We thank the reviewer for the suggestion to include additional references. These review articles summarise much of the content of the individual research articles that we have already cited within the introduction and discussion. We have added these to the manuscript (references 24 and 25).